# The Innate Immune Microenvironment in Metastatic Breast Cancer

**DOI:** 10.3390/jcm11205986

**Published:** 2022-10-11

**Authors:** Chiara Tommasi, Benedetta Pellegrino, Anna Diana, Marta Palafox Sancez, Michele Orditura, Mario Scartozzi, Antonino Musolino, Cinzia Solinas

**Affiliations:** 1Medical Oncology and Breast Unit, University Hospital of Parma, 43126 Parma, Italy; 2Department of Medicine and Surgery, University of Parma, 43126 Parma, Italy; 3GOIRC (Gruppo Oncologico Italiano di Ricerca Clinica), 43126 Parma, Italy; 4Medical Oncology Unit, Ospedale del Mare, 80147 Naples, Italy; 5Tumor Heterogeneity, Metastasis and Resistance Laboratory, University of Basel, 4001 Basel, Switzerland; 6Division of Medical Oncology, Department of Precision Medicine, University of Campania Luigi Vanvitelli, 80131 Naples, Italy; 7Medical Oncology Department, University of Cagliari, 09042 Cagliari, Italy

**Keywords:** innate immunity, metastatic breast cancer, tumor-immune microenvironment, tumor-associated macrophages, dendritic cells, tumor-associated neutrophiles

## Abstract

The immune system plays a fundamental role in neoplastic disease. In the era of immunotherapy, the adaptive immune response has been in the spotlight whereas the role of innate immunity in cancer development and progression is less known. The tumor microenvironment influences the terminal differentiation of innate immune cells, which can explicate their pro-tumor or anti-tumor effect. Different cells are able to recognize and eliminate no self and tumor cells: macrophages, natural killer cells, monocytes, dendritic cells, and neutrophils are, together with the elements of the complement system, the principal players of innate immunity in cancer development and evolution. Metastatic breast cancer is a heterogeneous disease from the stromal, immune, and biological point of view and requires deepened exploration to understand different patient outcomes. In this review, we summarize the evidence about the role of innate immunity in breast cancer metastatic sites and the potential targets for optimizing the innate response as a novel treatment opportunity.

## 1. Introduction: Immune Response and Metastatic Spread

Metastatic disease is the major cause of morbidity and mortality in breast cancer (BC) patients [1]. From the very early phases of tumor development, interactions between tumor cells and the immune environment are remarkably tight. The immune system is first able to recognize and eliminate malignant cells, and the sole cells that are able to escape from immune surveillance will survive and proliferate. Immune cells and soluble factors are also important players in the following phases of dormancy, when disseminated tumor cells (DTCs) remain in a latent state as micrometastases, that afterwards will undergo progression, invasiveness, and metastasis [2].

Metastatic organotropism of BC is directed to the bones, liver, lung, brain, and skin. Organ-specific tissue-resident stromal cells (e.g., fibroblasts and epithelial cells in lung, liver Kuppfer cells, and brain endothelial cells) play a crucial role in homing metastatic cells and preparing the pre-metastatic niche [3], where immune cells are recruited and are able to create a permissive growth environment before the arrival of tumor cells [4].

Myeloid cells such as tumor-associated macrophages (TAMs) and neutrophils (TANs) can promote metastatic spread through blood and lymphatic vessels via the production of matrix metalloproteinases (MMPs) involved in the degradation of extracellular substrates such as collagen. Cytokines, such as interleukin (IL)-1, tumor necrosis factor (TNF)-α, and IL-6, can also determine invasiveness and metastasis. Transforming growth factor-β (TGF-β) produced by TAMs, myeloid-derived suppressor cells (MDSCs), and cancer-associated fibroblasts (CAFs) are regulators of the epithelial–mesenchymal transition and metastasis. TNF-α and IL-6 sustain the survival of tumor cells that reach blood and lymphatic circulation. The increased levels of circulating cytokines in the serum of cancer patients are thought to increase the expression of adhesion molecules on the endothelium or in target organs and chemokine–chemokine receptor interactions are responsible for the guided migration of tumor cells to future specific target metastatic sites. Homing to specific sites is followed by extravasation into tissues, and the adaptation of tumor cells to a foreign environment through interactions with immune, inflammatory, and stromal cells of the new niche. After reaching the secondary organ site, metastatic cells can either proliferate or enter a dormant state. By creating a pre-metastatic niche [5], delivering site-specific chemo-attractants [6], and forming a favorable milieu [7,8,9], the tumor immune environment (TIM) plays a major role in determining whether tumor cells will progress towards clinically manifested metastases [10].

The aim of this work is to summarize the most important evidence describing the role played by the innate immune system in the phases of BC progression and metastases.

## 2. The Cells of Innate Immunity and Their Role According to the Site of Metastases

In physiological conditions, myeloid cells maintain homeostasis during the processes of tissue repair and remodeling. They are released in the bloodstream from hematopoietic stem cells. In the bone marrow microenvironment, a multistep process leads to the formation of common lymphoid progenitors and common myeloid progenitors (CMPs) [11]. Granulocyte/macrophage lineage-restricted progenitors (GMPs) are born from CMPs and are precursors of macrophages, dendritic cells, and granulocytes (basophils, eosinophils, and neutrophils) [11]. The terminally differentiated myeloid cells are essential to combat infections and in the scavenger process through antigen presentation [12].

In tumors, myeloid cells enhance tumor growth through the secretion of soluble factors, playing a role in the promotion of angiogenesis, invasion, and metastases. The release of factors by tumor and stromal cells from TIM generates MDSCs, which influence the adaptive immune response. On the other hand, TIM is able to convert terminally differentiated myeloid cells into potent immunosuppressive cells [13].

Malignant transformation, tumor vascularization, and neoplastic cell migration can be driven by bone marrow-derived cells (BMDCs).

Figure 1 shows the different cells of the innate immune system involved in the BC metastatic niche.

### 2.1. Monocytes (M-DSCs)

Monocyte-derived suppressor cells (M-DSCs) originate from hematopoietic stem cells and then localize within TIM [7]. Different signals are involved in the recruitment of circulating monocytes in tissues, where they can differentiate into monocyte-derived macrophages or monocyte-derived dendritic cells [14]. 

M-DSCs are able to suppress in vitro T cell activation [15]: they can influence innate and adaptive immune responses by depleting nutrients that are essential for lymphocytes, generating oxidative stress, influencing lymphocyte trafficking and viability, and activating and expanding regulatory T cells (Treg) [7] while suppressing CD8^+^ T cell activity [16]. The imbalance between host anti-tumor immunity and tumor tolerance is mediated by monocyte chemoattractant protein-1 (MCP-1) [17]. CD14^+^CD16^+^ monocytes are stimulated by MCP-1: they are elevated in the serum of BC patients and their levels are associated with the tumor size and stage [17]. 

Colony-stimulating factor-1 (CSF-1), released from invasive BC cells, induces monocytes’ secretion of chemokine C–X–C motif ligand 7 (CXCL7), which enhances the chemotaxis of monocytes in BC sites. Recruited monocytes into TMI enhance the invasive behavior of BC cells, resulting in the progression of tumor size and distant metastases [18]. 

The immunosuppressive intrahepatic environment restricts the endogenous anti-tumor immunity. In addition, liver M-DSCs expand in response to granulocyte-macrophage colony-stimulating factor (GM-CSF), suppressing anti-tumor immunity in BC liver metastases [19]. The majority of liver M-DSCs co-express GM-CSF receptor (GM-CSF-R), indoleamine 2,3-dioxygenase (IDO), and programmed death-ligand 1 (PD-L1): a reduction in IDO and PD-L1 expression has been observed through the GM-CSF or GM-CSF-R blockade or with the use of small-molecule inhibitors of Janus-activated kinase 2 (JAK2) and STAT3 [20].

CD137 is a member of the TNF receptor superfamily, and it was found to increase the adherence of monocytes, regulating the migration of monocytes/macrophages to TIM both in vitro and in vivo. Moreover, CD137 promoted their differentiation into osteoclasts, favoring the colonization of BC cells in the bone [21]. 

### 2.2. Macrophages

Macrophages are a group of tissue-resident myeloid cells derived from circulating or tissue-resident macrophages originating from Yolk sac precursor cells [13,22]. Resident macrophages from different tissues are specific and different according to the corresponding organ site [23]: for example, they produce TGF-β in the brain [24], PPAR-γ in the alveoli [25], and GM-CSF in the liver (where they are called Kupffer cells) [26]. 

TAMs act on primary tumor growth, the anti-tumor adaptive immune response, and angiogenesis, stromal remodeling, and metastatic genesis and evolution (Figure 2) [27]. Hypoxia, cytokines such as IL-4 and IL-13 (produced by T helper (Th)2 cells) or IL-10 (produced by Treg), metabolic products of tumor cells, and immune complexes may determine the functional phenotype of TAMs: they can be polarized within “classical” or “pro-inflammatory” M1 macrophages, which switch their metabolism towards enhanced anaerobic glycolysis, pentose phosphate pathway activation, and protein and fatty acid synthesis under the influence of interferon (IFN)-γ, NF-κB, STAT-1, and IRF-5. On the other hand, cytokines such as IL-4, IL-13, and MYC influence the development of “alternative” M2 macrophages, having pro-tumor activity, with angiogenesis induction. M2 polarization of TAMs can also be induced by other signals, such as the presence of immune complexes with or without lipopolysaccharide or IL-1, IL-10, and TGF-β [28]. CSF-1 and C-C motif ligand 2 (CCL2) are the most important factors involved in M2 polarization and are involved in the recruitment of TAMs in TIM [29]. 

Elevated macrophage CSF-1 levels are correlated with marked M2 macrophage infiltration in human metastatic BC [30]. In fact, metastasized primary BC had higher tumor epithelial and stromal expressions of CSF-1 (*p* < 0.001 and *p* = 0.002, respectively) and CSF-1R (both *p* = 0.03) compared to non-metastatic cancers [31]. A high expression of CSF-1/CSF-1R and a high density of TAMs and CD3^+^ T-lymphocytes create an immunosuppressive tumour milieu [32] that is related to tumoral immune escape through the inhibition of T lymphocytes and to BC progression [31]. 

Chemokine MCP-1 and CCL2 synthesis, produced by both tumor and stromal cells [33], mediates the recruitment of C-C chemokine receptor 2 (CCR2) monocytes (receptor for CCL2) and their subsequent differentiation into metastasis-associated macrophages (MAMs) [34,35].

Different soluble factors mediate the role of TAMs during cancer progression. 

Macrophage migration inhibitory factor (MIF) is a pluripotent cytokine that regulates the immune response in TIM. MIF is upregulated upon the infiltration and accumulation of TAMs and has been linked to tumor cell survival. A high level of MIF is found in M1-activated macrophages. In vitro, its reduction can increase the immunogenic capability of TAMs, with a significant increase in cytokines, such as IL-2 (lymphocyte T activator) and TNF-α, but also a higher expression of major histocompatibility complex (MHC)-II, with a significant role in antigen presentation and increase in T lymphocyte infiltration numbers and the tumor-reactive responses [36].

Raf kinase inhibitory protein (RKIP) inhibits tumor invasiveness. In a mouse xenograft model, RKIP expression in tumors markedly reduced the number and metastatic potential of infiltrating TAMs. TAMs isolated from nonmetastatic RKIP^+^ tumors exhibit a reduced ability to drive tumor cell invasion and decreased secretion of pro-metastatic factors relative to metastatic RKIP^−^ tumors. The expression of RKIP is often reduced in triple-negative BC (TNBC) [37]. 

In a humanized mouse model inoculated with MDA-MB-231 and T-474 cells, BC cells undergoing EMT closely interact with TAMs through the release of GM-CSF, which is able to activate them. TAMs produce CCL18, which stimulates EMT, thus leading to metastatic spread. Interestingly, high NF-κB activity in tumor cells was shown to not only be essential to their mesenchymal properties but also important for the ability to secrete activating TAM cytokines [38]. 

Granulocyte colony-stimulating factor (G-CSF) expression is significantly associated with CD163^+^ TAMs and with shorter overall survival in primary BC [39]. In the mesenchymal stem-like subtype MDA-MB-231 TN cell line, the secretion of high levels of G-CSF is involved in TAM polarization to the immunosuppressive HLA-DR^low^ phenotype, promoting the migration of BC cells through the secretion of TGF-α [40]. 

Of note, a negative correlation has been shown between the expression of the vitamin D receptor (VDR) and metastasis in BC. Co-culture of VDR-overexpressing tumor cells and a macrophage cell line demonstrated that overexpression of VDR alleviated the pro-metastatic effect of co-cultured macrophages on BC cells and abrogated the induction of EMT. Administration of an active vitamin D metabolite exerted similar antimetastatic effects in BC cells in vitro and in a mouse model of BC in vivo, with preservation of VDR [41].

In the metastatic phase, hypoxia is an essential passage of tumor cell resistance to immune effector lysis. In in vitro BC models, a hypoxia-inducible factor-1 (HIF-1)-dependent pathway increases the expression of the metalloproteinase ADAM10, which mediates the activation of resistance mechanisms in cellular tumoral lysis [42].

Table 1 summarizes the most important markers and cytokines acting on TAMs at the different BC metastatic sites.

Evidence suggests that the interaction between macrophages and T lymphocytes in nodes is essential for BC nodal involvement [43,44] and relapse-free survival [9]. A high TAM density in TME was significantly associated with poor prognosis, irrespective of TAM marker expression (CD68^+^ or CD163^+^, all *p* < 0.001). CD163 is a highly specific marker for M2-like macrophages and could be used as a marker with CD68 to detect and calculate the ratio of M1/M2 and its prognostic impact [45]. 

Several cancer types, including BC, have a high metastatic propensity to bone or bone tropism [46]. Reciprocal engagement between tumor cells and normal bone cells increases the bone remodeling activity, metastasis establishment, and progression. Receptor activator of nuclear factor kappa B (RANK) and its ligand (RANKL) plays an essential role in bone remodeling [47]. The “seed and soil” theory postulates that tumor cells produce several tumor-associated factors, including interleukins (IL-1β, IL-6, IL-8, IL-11, IL-17), macrophage inflammatory protein 1α, TNFα, parathyroid hormone-releasing protein (PTHrP), and prostaglandin E (PGE2). This process increases RANKL expression and induces a decrease in the osteoprotegerin (OPG) levels in the stromal and osteoblastic cells of the bone. The OPG ratio promotes osteolytic bone destruction and the release of calcium and growth factors, such as TGF-β and insulin-like growth factor (IGF), that are accumulated in bone reservoirs, which stimulates the proliferation of tumor cells and the production of more cancer-related factors to promote continuous bone destruction [48,49]. Bisphosphonate treatment inhibits bone osteoclasts, which could be a potential therapeutic target for patients with high-lysyl oxidase primary BC. Indeed, the enzyme increases bone resorption and thereby creates a metastatic niche for circulating tumor cells (CTCs) [50]. RANKL expression can also be induced in bone metastasis: directly by tumor cells or by stimuli present in the bone, such as αvβ3 integrin, CD44, TGFβ-dependent signaling [51], and the hypoxic environment [52]. In addition, tumor cells expressing RANK/RANKL undergo EMT and migration to bone without affecting bone reabsorption [53,54], whereas in endothelial cells, RANK induces angiogenesis, vascular permeability, and tumor cell extravasation [55]. In a PyMT mouse model, the expression of RANK in luminal BC led to the recruitment of TAMs and TANs, which inhibited T lymphocyte recruitment and/or activity [56]. CCL2 and CXCL12 are well-known key signaling pathways in BC bone metastases [57,58]. 

Murine models of BC treated with zoledronic acid showed a significant decrease in the size of lung metastases compared to the control, and immunohistological staining showed that zoledronic-acid-treated mice had impaired TAM recruitment and infiltration into the tumor stroma and reduced neo-vascularization [59]. Tumors from control mice had significantly more intracytoplasmic VEGF staining (in TAMs and tumor cells) compared to zoledronic-acid-treated mice. This correlated with a decrease in the TAM density and serum VEGF levels. In addition, TAMs isolated from treated mice expressed inducible nitric oxide synthase (iNOS), a hallmark protein of M1 polarization, whereas control mice did not, suggesting that TAMs are a potential immune target of zoledronic acid therapy [59]. 

Hormone receptor-negative BC has a higher propensity to metastasize to lungs if the vascular cell adhesion protein (VCAM)-1 is expressed. In an in vitro model of single-cell suspensions prepared from lung metastatic nodules, VCAM-1 was able to bind to MAMs, the most abundant source of potential α4-integrin, and vascular endothelial cells but not to neutrophils or tumor cells. Of note, VCAM-1-expressing tumor cells had a survival advantage in metastatic sites usually rich in leukocytes, such as the lungs [60]. MAMs differ from CD11c^+^ lung interstitial resident macrophages: they are regulated by CSF-1 and characterized by cell surface expression of CD11b, vascular endothelial growth factor receptor 1 (VEGF-R1), and CCR2 [61]. The ablation or deletion of CCL3 inhibits metastatic lung seeding and metastases growth [34,61,62]. 

Microglia, resident macrophages in the brain, have been shown to play a prominent role in metastasis formation, enhancing invasion and colonization by BC cells through in a JNK-dependent way. Pro-invasive microglia with altered morphology colocalized with tumor cells, without upregulation of M2-like cytokines nor differential gene expression after co-culture with MCF-7 BC cell lines. In an organotypic slice coculture model, microglial cells are able to transport tumor cells into brain tissues and its inactivation inhibited malignant invasion in living brain tissue. Already invaded tumor cells were always preceded by microglia in the infiltration zone, suggesting that these macrophages not only act as guiding rails but actively prepare the way for invasion and colonization [63]. 

### 2.3. Dendritic Cells (DCs) 

DCs are professional antigen-presenting cells (APCs) that can positively or negatively influence the adaptive immune response [64]. As terminally differentiated myeloid cells, DCs specialize in antigen processing and presentation and monocytes are their major precursors in humans. These cells reside in tissues in an immature, non-active state [7]. They become activated and undergo maturation in response to stimuli associated with bacteria, viruses, and tissue damage [7]. Only functional activated DCs are able to stimulate an effective T cell response. In cancer, DCs undergo abnormal differentiation, with decreased production of mature functionally competent DCs, increased accumulation of immature DCs at the tumor site, and increased production of immature myeloid cells [65]. DCs and macrophages first recognize and bind to the dying BC cells or release tumor-associated antigens through pattern recognition receptors (PRRs) [66]. PRRs can identify and recognize the damage-associated molecular patterns (DAMPs), which are derived from the tumor or dying cells to drive intrinsic tumor inflammation [66]. 

GM-CSF with IL-4 is a potent growth factor for DCs [67]. BC-derived GM-CSF has a pro-tumor role and high levels of endogenous GM-CSF are associated with metastasis, progression, and reduced survival in patients with BC [68]. On the other hand, patients treated with neoadjuvant chemotherapy and exogenous GM-CSF showed a significantly higher mean percentage of DCs in TIM, with a longer disease-free survival [69]. 

RANKL augments the ability of DCs to stimulate naïve T lymphocyte proliferation [70], whereas activated T lymphocytes that express RANKL enhance the survival of DCs, increasing inflammation [71,72]. On the other hand, RANKL induced in keratinocytes can regulate the activation of DCs and induce immunosuppression, which is crucial for the peripheral homeostasis of Tregs [73]. 

Hypoxia, the accumulation of extracellular adenosine, increased levels of lactate, and a decreased pH in TIM can affect DC migration and function [74]. Resident conventional DCs confer anti-metastatic protection in the healthy lung tissue. CCR2-knockout mice develop fewer lung metastases from primary BC, with higher cytoplast loading by host-protective CD103^+^ DCs and a higher frequency and number of CD8^+^ T lymphocytes. In this way, DCs act in competition with pro-tumor macrophages and have also been shown to play a role in limiting metastases [75].

### 2.4. Tumor-Associated Neutrophils (TANs)

Neutrophils are indispensable antagonists of microbial infection and facilitators of wound healing. The traditionally held belief that neutrophils are inert bystanders is being challenged by the recent literature [76]. The presence of granulocytes, particularly neutrophils, has been linked with tumor angiogenesis and metastases [77]. Tumors may polarize neutrophil phenotypes during tumor progression, resulting in either tumor destruction or survival at metastatic sites [77]. 

The precise role of neutrophils in metastasis remains uncertain [77]. In fact, neutrophils can have dichotomic polarization, being able to shift from an anti-tumor (N1) to a pro-tumor (N2) profile [78]. TGF-β in TIM mediates the transformation between the N1, which involves pro-inflammatory neutrophiles with the capacity to stimulate effector T lymphocytes, and the N2 phenotype, which has pro-tumor activity with immunosuppressive and angiogenic features [79]. Of interest, the tumor-promoting activity of TANs can be reversed to an anti-tumor role with TGF-β blockade [80]. 

N1 neutrophils can exert anti-tumor functions through an antibody-dependent cellular cytotoxicity (ADCC) effect, producing radical oxygen species (ROS), TNF-α, and nitric oxide with a direct killing effect, and inhibiting suppressive cells, such as IL-17-producing γδ T lymphocytes. To the contrary, N2 can produce CCL2 and CCL17 to recruit CD4^+^ T cells and anti-inflammatory macrophages together with arginase-1 to inhibit CD8^+^ T lymphocyte activation, promoting an immunosuppressive TIM. They also promote tumor angiogenesis, releasing MMP9 and VEGF, and promote tumor cell proliferation and EMT via IL-6, IL-1β, and IL-17 release [81]. 

Upon arrival in the pre-metastatic niche, BMDCs secrete factors that facilitate tumor cell survival and growth [6,82]. In a preclinical BC study, TANs were the predominant population in the early/pre-metastatic lung, and their depletion reduced metastases to the lung [82]. In a mice model, the administration of G-CSF increased neutrophil recruitment and accumulation in primary tumors and blood, leading to an increased metastatic capacity and reduced survival [82].

In a mouse model of BC lung metastases, an abundancy of immature Gr-1^+^CD11b^+^ myeloid cells was observed in the lungs before the arrival of tumor cells. These cells shifted from anti-tumor IFN-γ production to an increase in pro-inflammatory cytokine production, such as MMP9, promoting aberrant neo-angiogenesis [83]. Tumor and tumor-associated stromal cells produce neutrophil-attracting CXC-chemokines and prokineticin 2 [7]. In the lungs, tumor-derived G-CSF also mobilizes granulocytes to pre-metastatic niches and supports subsequent metastasis formation, whereas prokineticin-2 aids tumor cell migration through the activation of prokineticin receptor [82]. TANs are able to promote angiogenesis in primary tumors and metastatic sites. Gr-1^+^ cells were able to produce MMP-9 and promote vascular remodeling in the pre-metastatic niche. On the other hand, neutrophils promote the establishment of tumor cells in the lung via the induction of MMP-9-mediated angiogenesis [83].

N2-neutrophils promote the release of STAT3-activated lipocalin 2 (LCN2), a secretory glycoprotein, and induce EMT, thereby facilitating colonization and metastatic outgrowth. The levels of LCN2 in serum and saliva are elevated in early stage BC patients and cancer-free females with a history of smoking, suggesting that LCN2 serves as a promising prognostic biomarker for predicting increased risk of metastatic disease in female smokers [84]. 

The neutrophil to lymphocyte ratio (NLR), calculated as the neutrophil count divided by the lymphocyte count, is consistently reported as an unfavorable prognostic indicator for patients with gastrointestinal, lung, renal, and gynecological cancers [85]. One potential mechanism underlying the prognostic impact of high NLR with poor outcomes may be associated with systemic inflammation: elevated circulating concentrations of cytokines (such as IL-1, IL-6, IL-12, IFN-γ, MCP-1) are associated with systemic inflammation and neutrophilia inhibits the cytolytic activity of T lymphocytes [86]. Neutrophils secrete tumor growth factors, including VEGF and hepatocyte growth factor (HGF), but also MMPs and elastases, and thus likely contribute to a pro-tumor TIM [86]. A high NLR represents an easily measurable and inexpensive marker of systemic inflammation [86] and is associated with an adverse overall survival and disease-free survival in patients with BC, and its prognostic value is consistent among different clinicopathologic factors such as disease stage and subtype [85,87].

### 2.5. Mast Cells and Natural Killer (NK) Cells

Mast cells are derived from hematopoietic stem cells. They secrete cytokines that are involved in T lymphocyte responses. In addition, mast cells are able to influence natural killer (NK) activity through the release of granzyme B [88]. They also play a role in tissue remodeling by releasing enzymes in the microenvironment and by interacting with fibroblasts and myofibroblasts. They secrete TGF-β1, an enhancer of fibrogenesis and extracellular matrix production, and proteases activate MMPs [89]. A high mast cell density has been correlated with lymph-node metastases [90,91]. Mediators released by mast cells (histamine, TNF, VEGF, and tryptase) can increase vascular permeability, enhancing the extravasation and metastatic spread by tumor cells [92]. In a BC cell line model, human mast cells were shown to enhance the invasive property of tumor cells through the HLA-G–KIR2DL4 axis [88]. 

Among the cells of the innate immunity, NK cells are able to recognize and kill tumor cells expressing stress-ligands and non-expressing MHC-I on their surface. NK are activated by MHC-I-negative cells, priming local DCs and stimulating a strong protective response by CD8^+^ T lymphocytes. Given that the disseminated metastatic cells recovered MHC-I cell surface expression, they might be recognized and kept in dormancy by CD8^+^ T lymphocytes [2].

NK cells play different roles in the various stages of tumor development [93]. In the primary tumor, they can promote potent anti-tumor functions and can be inhibited by M-MDSCs and Tregs. In peripheral blood, they are able to recognize and kill DTCs that are not coated by platelets. In the pre-metastatic niche, NK cells can be part of tumor-infiltrating leukocytes before CTCs seeding; in metastatic lesions, NK cells can be suppressed by IL-10, TGF-β, and adenosine, leading to increased tumor growth [93]. NK cells can also induce the activation of DCs, stimulating the adaptive immune response. These cells are also involved in keeping a check on DTCs during the phases of tumor dormancy. In bone marrow from BC patients, DTCs and several immune subpopulations, including NK cells, macrophages, and T lymphocytes, were observed. They had increased expression of markers of activation, proliferation, co-stimulation, and memory [94].

The percentages of conventional regulatory NK cells in BC tissue were positively correlated with the tumor size (higher percentages in T3 compared with smaller T1) [95]. The percentages of NK cells expressing activation markers such as NKG2A, CXCR3, Granzyme B, and Perforin, were not significantly different between patients based on the clinicopathological characteristics and different BC phenotypes [95]. The accumulation of NK cells and the expression of activating NKG2D receptor by tumor-infiltrating NK cells may play roles in BC regression. Indeed, NKG2D was expressed in about half of the NK cells accumulated at the site of tumor and was observed to be more frequent in node-negative BC patients [95].

In an immunocompetent BALB/c mice model, the rupture of the balance between NK cells and hepatic stellate cells (HSCs) results in the reversal of dormancy of the BC milieu in the liver. Increased levels of IL-15 induce the proliferation of NK cells, and the dormancy of BC cells is achieved through IFN-γ-induced quiescence. The activation of HSCs and the secretion of CXCL12 act on CXCR4 in NK cells and determines their quiescency. CXCL12 expression and HSC abundance are closely correlated in patients with liver metastases, mirroring the interplay between the immune response and the hepatic microenvironment [96].

In tumor-bearing immunocompetent mice, NK cells may promote the development of a cytotoxic immune response, independent of CD4^+^ T lymphocytes, as the depletion of CD8^+^ T lymphocytes promoted the onset of lung metastases [2].

In a mice model, the administration of an antibody targeting CD96 in NK cells protected against the experimental development of lung metastases, and this repression required the presence of NK and IFN-γ [97]. Of note, a combination of an anti-CD96 with the cytotoxic T lymphocyte-associated protein-4 (CTLA-4) or with the anti-programmed cell death protein-1 (PD-1) immune checkpoint-blocking agents showed an anti-metastatic activity. Particularly, anti-PD-1 in association with anti-CD96 increased the function of lung NK cells, leading to tumor regression. NK cells were critical for the anti-tumor activity of this combination but not T lymphocytes, as shown by the effects exerted by the depletion of CD4^+^ and CD8^+^ T lymphocytes [97].

In addition, in a mouse model of BC brain metastases, the administration of EGFR-CAR NK cells alone or in combination with an oncolytic herpes virus-1 resulted in more efficient eradication of tumor cells in vitro and more efficient killing of MDA-MB-231 tumor cells in an intracranial model [98].

### 2.6. Complement System

The complement system is a cascade of serine proteases encoded by genes originating from the same ancestral genes as coagulation proteins. Its activation involves several steps and is tightly regulated. Many complement proteins possess dual functions that provide crosstalk between the complement system and other effector and regulatory systems. As a result, the complement system participates in adaptive immunity, hemostasis, and organ development in addition to its role in innate immunity [99]. The extracellular body compartment is the main environment for the activation of the plasmatic complement system cascade [100].

The complement system is known to play a dual role in cancer [101]. As a fundamental part of the innate immunity, it is capable of targeting tumor cells and managing the immune response against the tumor [101]. On the other hand, as a potent pro-inflammatory mechanism, the complement system is thought to substantially contribute to tumor growth by generating a chronic inflammation state that facilitates mobilization of immune suppressor cells and supports angiogenesis [101].

In BC, local expression of complement inhibitors was reported as a mechanism of evading cytotoxic complement function [100]. In primary BC, the expression of factor I of complement and CD46 correlated with a larger tumor size, higher grade, and poor prognosis [100]. Moreover, in animal models of BC, the complement system has a role in lung premetastatic niche formation [102]. Anaphylatoxin C5a, released from C5 by tumor cells, binds to C5a receptor (C5aR) and acts as a leukocyte chemoattractant and inflammatory mediator. C5aR expression in BC is associated with poor prognosis and more extensive nodal involvement [103].

Using two murine BC models (EMT6 and 4T1), treatment with a dual C3aR/C5aR1 agonist significantly slowed mammary tumor development and progression, suggesting that complement activation peptides can influence the anti-tumor response in different ways [104].

Over-sulfated glycosaminoglycans (GAGs) induce thrombin generation through contact system activation. Plasma from BC patients contains activated contact systems for the absence of high-molecular-weight kininogen and processed C1-inh (molecules of the complement system), abnormal kallikrein and thrombin activities, and increased glucosamine, galactosamine, and GAG levels. These data suggest that GAGs or other molecules produced by tumors induce abnormal thrombin generation through contact system activation, resulting in the hypercoagulable state of cancer patients [105,106].

## 3. The Development of Pre-Metastatic Niches

The identification of mechanisms by which pre-metastatic niches contribute to tumor outgrowth is an area of active investigation. However, little is known about the influence of extrinsic environmental factor(s) on organ-specific metastatic progression against a background of massive attrition of DTCs [84].

The bone marrow-derived hematopoietic progenitors are mobilized in response to the array of growth factors produced by the primary tumor. Their arrival in distant sites represents early changes in the local microenvironment, termed the ‘‘pre-metastatic niche’,’ which dictates the pattern of metastatic spread [4].

A particular subset of VEGFR1^+^ hematopoietic BMDCs, located in specified niches in the bone marrow, are able to proliferate and mobilize to the bloodstream during the phases of tumor development [6]. These cells localize to perivascular sites, stabilizing tumor neo-vessels and colonizing metastatic sites before tumor cells, generating the pre-metastatic niche [4]. In humans, VEGFR1^+^ clusters were observed in both primary tumors (including BC) and secondary lesions [6]. Of interest, a higher density of cellular clusters was observed before tumor spread in sites of metastasis [6]. Chemoattraction and attachment of BC, lung cancer, and melanoma CTCs were promoted by VEGFR1^+^ progenitors with the aid of fibroblasts, fibronectin, and the CXCL12/CXCR4 axis [6].

High levels of VEGFR2^+^ endothelial BMDCs are present in the blood of patients with metastatic pediatric solid malignancies [107]. In BC patients, VEGFR1^+^ hematopoietic BMDCs increased in the months before relapse followed by a surge in VEGFR2^+^ endothelial BMDCs immediately preceding an overt relapse of cancer. In this view, VEGFR1^+^ cells represent initiation of the pre-metastatic niche, and a manifest relapse occurs after VEGFR2^+^ cells start the pro-angiogenic switch [108].

Tumor-derived growth factors such as chemokines and cytokines facilitate recruitment of M-MDSCs and TAMs into tumors. CCL2 is associated with poor prognosis in BC [109]. Through recruitment of CCR2-expressing M-MDSCs, CCL2 has been shown to promote pulmonary metastasis in mouse models of BC. Activation of the CCL2–CCR2 axis promotes CCL3 production from macrophages, enhancing metastatic seeding of breast tumor cells [109]. M-MDSCs and TAMs expressing CD11b are supposed to play a key role in the development of lung metastasis from primary BC. They are recruited from blood circulation during the formation of the pre-metastatic niche. In contrast, resident conventional DCs confer anti-metastatic protection [75]. During metastatic BC cells’ seeding, CTCs are detectable in the blood of patients. Once these cells reached the lung microvasculature in a mouse model, they began to shed microparticles into the vasculature. In pre-metastatic mice lungs, the fragments called cytoplasts, with metabolic potential and motility, were the primary source of ingested tumor material in the phagocytosis process exerted by myeloid cells [75]. Many of these tumor-ingesting myeloid cells accumulated in the lung interstitium along with metastatic cells, promoting the development of lung metastases [75].

## 4. Therapeutic Implications

Escape of BC from immunosurveillance often results from diminished effector immune cell function and the immunosuppressive TIM. Therefore, therapies targeting the innate immune system may represent a promising next-generation approach for patients with BC [110]. Table 2 summarizes the available clinical trials targeting the innate immune system. 

Stimulator of interferon genes (STING) is an intracellular DNA recognition receptor that can induce type I IFN production and host innate immune activation [111]. Cyclic dinucleotides, such as cyclic guanosine monophosphate–adenosine monophosphate (cGAMP), have been shown to improve vaccination in multiple cancer types, including BC [111]. Low-dose cGAMP significantly increased the production of IL-12 by MDSCs, with improved T lymphocyte responses to MAGE-b, whereas a high dose of cGAMP activated caspase-3 in the 4T1 tumor cells and killed the tumor cells directly. The activation of STING-dependent pathways by cGAMP is highly attractive for cancer immunotherapy [112]. A phase I clinical trial tested the efficacy of MIW815 (ADU-S100), an intratumoral STING agonist, showing clinical activity in terms of stability and partial response in 94% of treated patients. However the increase in inflammatory cytokines and peripheral blood clonal T lymphocytes suggested a systemic immune activation [113] and represents one of the major limitations in the formulation of systemic use of STING agonists. 

CSF-1R is crucial for the differentiation and survival of the mononuclear phagocyte system and in particular the M2 polarization of TAMs. Different drugs directed at CSF-1R or its ligand are in clinical development as monotherapy or as combination treatments [114]. The small molecule ARRY-382 [114] and the monoclonal antibody Emactuzumab [115] showed a clinical benefit rate of 15% and 24%, respectively, as a monotherapy treatment of metastatic cancer patients. Other CSF-1R inhibitors are in development in combination with anti-PD-1 agents [116]. 

D2 dopamine receptor (DRD2) belongs to the dopamine receptors family, and regulates the re-programming of TIM [117]. It facilitates M1 polarization of macrophages, inhibits the NF-κB signaling pathway, and triggers different processes of programmed cell death in BC [117]. In BC, higher expression of DRD2 is positively correlated with longer survival times, especially in the HER2-positive subtype. DRD2 also promoted BC cell sensitivity to chemotherapy [117]. ONC201 is a small molecule that selectively inhibits DRD2, resulting in activation of the integrated stress response combined with inactivation of Akt/ERK and other pro-survival signaling pathways [118]. A first-in-man phase I single-agent tested the efficacy and tolerability of oral ONC201 in patients with advanced cancer, including BC [119]. 

TAMs recognize and engulf tumor cells, a process termed “programmed cell removal” (PrCR), an efficient process of cancer immunosurveillance that can be initiated independently of the induction of tumor cell death [120]. CD47 was reported to be highly expressed in TNBC and is associated with a worse prognosis and outcome [121]. In pre-clinical models of TNBC, the blockade of CD47 was sufficient in inducing PrCR of TNBC cells, but its efficacy was not satisfactory [121]. On the other hand, the combination of a CD47 blockage with cabazitaxel, a microtubule depolymerization inhibitor [122], elicited a strong anti-tumor effect, inducing polarization of TAMs towards an M1 state by the activation of TLR/NF-kB pathways and enhancing the expression of pro-inflammatory cytokines. This study suggests a novel therapeutic opportunity in TNBC patients [121]. NCT04349969 is also testing an anti-CD47 antibody in monotherapy with promising results for further combinations with anti-PD-1 or anti-CTLA-4 antibodies [123]. 

Adjuvants are derived from DAMPs to improve the potency of BC treatment. They can stimulate innate immune cells, such as DCs, by binding to the expressed PRRs, including toll-like receptors (TLRs) and STING-dependent DNA sensors [111]. Poly (I: C), a ligand for TLR3, modulated MDSC activity [124]. Poly (I: C) directly bonded to TLR3 expressed on the surface of MDSCs, causing a reduction in circulating and infiltrating MDSCs in BC. The use of R484 alone has been demonstrated to delay tumor growth in BC models [125]. The I-SPY2 trial tested a TLR9 agonist (SD-101) combined with the anti-PD-1 pembrolizumab in the treatment of HER2-negative BC to investigate this novel anti-tumor and immunotherapeutic strategy [126]. OX40 is a transmembrane glycoprotein that is expressed on both activated CD4+ and CD8+ T cells, neutrophiles, and NKs. OX40 can strongly promote the induction of CD4^+^ T helper lymphocytes, and under other conditions, it can inhibit the generation of Foxp3^+^ Treg [127]. A clinical trial evaluating the efficacy and safety of intra-tumoral injection of a TLR9 agonist in combination with intra-tumoral and intravenous anti-OX40 antibody is ongoing [128].

CD137 is a co-stimulatory molecule belonging to the TNF receptor superfamily involved in cell differentiation. In addition, it is involved in the production of several inflammatory cytokines such as IL-6, TNF-α, and MCP-1 in adipocytes and macrophages. Anti-CD137 antibody therapy has been shown to severely deplete CD4^+^, B lymphocytes, and NK cells [129]. The NCT04648202 clinical trial is the first-in-man study evaluating the activity of FS120, an OX40/CD137 bispecific antibody, both in monotherapy and in combination with pembrolizumab in patients with advanced malignancies [130].

NK cells express classical checkpoint receptors, including PD-1 [131], CTLA-4 [132], and LAG-3 [133], and they have the potential to improve anti-PD-L1 monoclonal antibody efficacy through the antibody-dependent cellular cytotoxicity (ADCC) [134]. The use of anti-NKG2A (monalizumab) [135,136] or anti-KIR (lirilumab) antibodies seems to reconstitute the anti-tumor NK cell cytotoxic response [134]. 

## 5. Conclusions

Although the treatment of solid tumors has been revolutionized by the introduction of immunotherapy (mostly with immune checkpoint blockade) into clinical practice, its application for the treatment of metastatic BC appears limited compared to other malignancies (i.e., melanoma or lung cancer). The lessons learnt from early phase trials revealed that PD-(L)1 checkpoint blockade is more active in the first-line setting and in PD-L1-positive metastatic TNBC [138].

In the metastatic setting, it is important to study the role of innate immunity considering the different actors that play a role in all the phases of the disease, from the homing of the pre-metastatic niche to the treatment-resistant phase. The evaluation of new therapeutic strategies to manipulate innate immunity represents a challenge and a possible new approach for the treatment of metastatic BC patients.

## Figures and Tables

**Figure 1 jcm-11-05986-f001:**
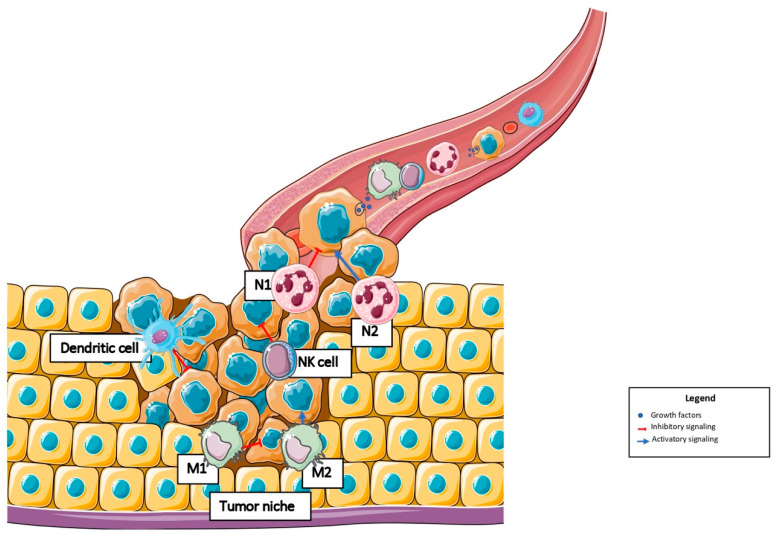
The role of the tumor microenvironment in the composition of the tumor niche in breast cancer. Abbreviations: N1: tumor-associated neutrophil type 1; N2: tumor-associated neutrophil type 2; M1: tumor-associated macrophage type 1; M2: tumor-associated macrophage type 2; NK cell: natural killer cell.

**Figure 2 jcm-11-05986-f002:**
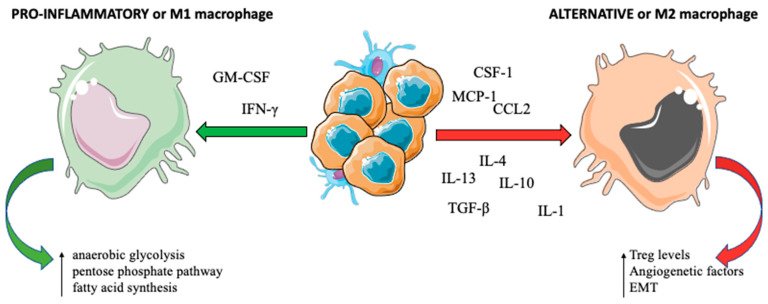
Cytokines and chemokines involved in macrophagic polarization. Tumoral and stromal cells secrete growth factors involved in monocyte attraction and macrophagic differentiation in M1 pro-inflammatory macrophages, with anti-tumor function, and M2 alternative macrophages, with pro-apoptotic activity.

**Table 1 jcm-11-05986-t001:** Role of tumor-associated macrophages (TAMs) at different breast cancer (BC) metastatic sites.

Metastatic Site	Marker and Cytokines	Distribution and Function in Healthy Tissue	Role in BC Metastases	Ref.
Nodes	CD168	Macrophages of marginal and medullary sinuses	↓CD163^+^ TAMs and ↑Foxp3^+^ Treg in metastatic SLN	[43]
	CD68	Monocyte lineage and resident macrophages	high CD68/(CD3+CD20+) ratio in BC primary tumors is associated with shorter RFS	[9]
			CD68+/stabilin-1+ cells in BC stroma are directly correlated with the number of metastatic nodes	[44]
	CD163	Resident macrophages	M2 marker; high levels are associated with bad prognosis	[45]
Bone	CD137	Circulating monocytes	Increases monocyte adhesion and induces osteoclast differentiation, favoring BC cell bone homing	[21]
	CCL2	Mediates the recruitment of monocytes and T lymphocytes	Induces macrophage-mediated bone destruction and tumor cell proliferation through VEGF-induced pathways	[57]
	CXCL12	On different tissues (brain, thymus, lung, liver) with chemotactic properties	Facilitates BC cells bone homing	[58]
Lung	VCAM-1	On endothelial cells stimulated with cytokines; it binds to α4β1-integrin expressed by monocytes	VCAM-1/α4β1-integrin interaction is determinant for tumor cell extravasation and migration through the pulmonary endothelium	[60]
	CD11b/CCR2^+^/VEGFR1^+^	Monocytes	Combined expression induced by cytokines, facilitates lung metastatic involvement	[61]
	CCL3	Lung-resident macrophages	In vivo model, its deletion or depletion of its receptor CCR1 in MAMs, reduce the number of lung metastasis foci	[62]

Abbreviations: TAMs: tumor-associated macrophages; BC: breast cancer; SLN: sentinel lymph node; RFS: relapse free survival; VEGF: vascular endothelial growth factor; VCAM-1: vascular cell adhesion protein-1; CCR1: C-C chemokine receptor 1; MAMs: metastasis-associated macrophages.

**Table 2 jcm-11-05986-t002:** Available compounds targeting the innate immunity system.

Target Cells	Compound	Effect	Administration	Trial Phase	Ref.
MDSCs	STING agonist	Increases the production of IL-12 with induction of the T lymphocyte response	IT	I	[113]
TAMs	ARRY-382 (CSF-1R inhibitor)	Reduces T-lymphocyte-suppressive TAM infiltrates	PO	I	[114]
	Emactuzumab (anti-CSF-1R)		IV	I	[115]
	Pexidartinib (CSF-1R inhibitor)	In combination with pembrolizumab, it inhibits M2 TAM polarization and restores the immune response against tumor cells	PO	I/II	[116]
	DRD2 antagonist	Increases M1 TAM polarization	PO	I	[119]
	CD47 antagonist	Induction of M1 TAM polarization and PrCR	IV	I	[123]
NKs	TLR9 agonist	In combination with anti-OX40, it enhances the activity of immune cells against tumor cells	IT	I	[128]
	FS120 (anti-OX40/CD137)	As single agents or in combination with pembrolizumab, it activates the cytotoxicity of CD8^+^ T lymphocytes and NKs and reprograms Tregs	IV	I	[130]
	Monalizumab (anti-NKG2A)	In combination with trastuzumab in metastatic HER2-positive BC, it improves ADCC and overcomes trastuzumab resistance. In addition, it can promote anti-tumor immunity by unleashing NK cells and CD8^+^ T lymphocytes	IV	II	[135,136]
	Lirilumab (anti-KIR)	In combination with nivolumab, it enhances NK cytotoxicity	IV	I	[137]

Abbreviations: MDSCs: myeloid-derived suppressor cells, STING: stimulator of interferon genes; IT: intratumoral injection; TAMs: tumor-associated macrophages; DRD2: D2 dopamine receptor; PO: per os; NKs: natural killers; PrCR: programmed cell removal; TLR9: toll-like receptor 9; IV: intravenous; ADCC: antibody-dependent cellular cytotoxicity.

## Data Availability

Not applicable.

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
