# Peer review of "The Innate Immune Microenvironment in Metastatic Breast Cancer"

_jcm, 2022, doi:10.3390/jcm11205986_

Round 1

Reviewer 1 Report

Dear Authors:

The manuscript by Tommasi et al has summarized the evidences about the impact of innate immunity on breast cancer metastatic sites and the potential targets to optimize the innate response as a novel treatment opportunity. I have just a few suggestions.

1. Some background information and references are missing:

Some other reviews also demonstrated the association between TAM and breast cancer. (please cite: 1.https://www.frontiersin.org/articles/10.3389/fonc.2022.820968/abstract Front. Oncol., 22 June 2022 Sec.Breast Cancer https://doi.org/10.3389/fonc.2022.820968 

2.Mitochondrial mutations and mitoepigenetics

2. If it is possible, please add more figures, like potential mechanism figure about TAM and breast cancer.

Best,

Author Response

Thanks for the useful advises!

Response 1 and 3:

We have revised the manuscript and added new and updated references in all the paragraphs as you suggested! in addition, we created new tables and figures about the role of TAMs and the  therapeutic implications.  

Response 2:

Although it is a very interesting topic, we didn't include mitochondrial mutations and mitoepigeneics part for the lack of evidences related to the impact of them in the innate immunity against breast cancer metastases. 

Reviewer 2 Report

This manuscript summarizes the innate immunity on breast cancer and the treatment opportunity, which includes important topics.

I have one major and some minor comments.

Major:

1 , Angiogenesis has a critical role on the tumor microenvironment and treatment strategy. I suggest the authors add a section and a figure about immunity and angiogenesis, which would review about the relation of immune cells and VEGF, VEGFR1, VEGFR2, Angiopoietin, Apelin or other factors. 

Minor:

2, Section 2.1. Macrophages includes a lot of topics in about 2 pages, which is long to follow. I suggest the section 2.1 be divided into some sections. 

3, In section 3. describes the relation of VEGFR1+ cells and pre-metastatic niches. I suggest the authors mention VEGFR2+ cells in this context, if possible.

Author Response

Thanks for the comments and advises!

We modified and added informations as requested for the minor comments: in particular we shortened the TAMs' paragraph and we added a figure and a table for it. In addition we have explained also the role of VEGFR2+ cells. 

To response to the major comment, we didn't debate the role of angiogenesis in breast cancer because of the lack of evidences and efficacy of antiangiogenic treatments in this context. 

Reviewer 3 Report

This review “The innate immune microenvironment in metastatic breast cancer” by Tommasi C et al.  can be interesting, but needs to be improved. All the comments given below should be addressed.

1)In Introduction authors should discuss what are myeloid cells, their classification and their functions in healthy tissue and in cancer.

2)Figure 1 doesn`t show any mechanism, it is just cells on it. Figure should reflect the main mechanisms realized by cells of innate immunity in metastasis.

3)The review lacks structure, there are no logical transitions between different topics. For example, in the chapter on macrophages, the authors first talk about the MIF factor, then about hormone-dependent breast cancer, Raf kinase inhibitory protein (RKIP), CD163, also about bone marrow, and so on. The structure should be as follows: start with progenitor cells (monocytes), their recruitment from the bone marrow, then macrophage origin followed by the role of CSF-1 and MCP-1 in progression, after that indicate the key factors secreted and expressed during each tumor process (angiogenesis, invasion and migration of tumor cells, metastases and formation of metastatic niches, etc.). Conclude with correlations of the main macrophage markers with metastasis of the tumor (for example, as shown in Front Oncol. 2020 Oct 22;10:566511. doi: 10.3389/fonc.2020.566511, J Hematol Oncol 12, 76 (2019). https://doi.org/10.1186/s13045-019-0760-3).

4)For tumor immune environment it is best to introduce an abbreviation “TIM”.

5)It is not clear the discussion of M-MDSCs in dendritic cell part.

6)The part “Therapeutic implications” should contain only studies on the anti-metastatic effect of developing therapeutic agents.

7) More most relevant references is needed.

8) Reviewer recommend adding the table, including the information of major markers of innate immune cells involved in metastasis in breast cancer cohorts of patients.

8) Finally, the major comment - if the goal set by the authors was that of considering the role of cells in metastasis, then more emphasis should be placed on the role of all immune cells in metastasis, this should be the leading thread of the narrative.

9) English editing is needed.

Author Response

Thanks for the useful recommendations! 

1)In Introduction authors should discuss what are myeloid cells, their classification and their functions in healthy tissue and in cancer.

Response 1: we have implemented the introduction with requested informations.

2)Figure 1 doesn`t show any mechanism, it is just cells on it. Figure should reflect the main mechanisms realized by cells of innate immunity in metastasis.

Response 2: the figure shows all the cells implicated in the innate immunity. The most important role about activation/inhibition signals are included in the legend. 

3)The review lacks structure, there are no logical transitions between different topics. For example, in the chapter on macrophages, the authors first talk about the MIF factor, then about hormone-dependent breast cancer, Raf kinase inhibitory protein (RKIP), CD163, also about bone marrow, and so on. The structure should be as follows: start with progenitor cells (monocytes), their recruitment from the bone marrow, then macrophage origin followed by the role of CSF-1 and MCP-1 in progression, after that indicate the key factors secreted and expressed during each tumor process (angiogenesis, invasion and migration of tumor cells, metastases and formation of metastatic niches, etc.). Conclude with correlations of the main macrophage markers with metastasis of the tumor (for example, as shown in Front Oncol. 2020 Oct 22;10:566511. doi: 10.3389/fonc.2020.566511, J Hematol Oncol 12, 76 (2019). https://doi.org/10.1186/s13045-019-0760-3).

4)For tumor immune environment it is best to introduce an abbreviation “TIM”.

5)It is not clear the discussion of M-MDSCs in dendritic cell part.

6)The part “Therapeutic implications” should contain only studies on the anti-metastatic effect of developing therapeutic agents.

Responde 3, 4, 5 and 6: Thanks for the suggestion! We have completely changed the paragraphs following this advices. 

7) More most relevant references is needed.

8) Reviewer recommend adding the table, including the information of major markers of innate immune cells involved in metastasis in breast cancer cohorts of patients.

Response 7 and 8: we have added a lot of new and updated references and we have created new tables and figures. 

9) Finally, the major comment - if the goal set by the authors was that of considering the role of cells in metastasis, then more emphasis should be placed on the role of all immune cells in metastasis, this should be the leading thread of the narrative.

Response 9: the old manuscript lost the leading thread of the manuscript focused on innate immunity on metastases, so we modified the paper to return to the original idea. 

Round 2

Reviewer 1 Report

no comment

Reviewer 2 Report

The authors well modified the manuscript in accordance with the reviewers' comments.